# RingFFL: A Ring-Architecture-Based Fair Federated Learning Framework

Lu Han [1], Xiaohong Huang [1,*], Dandan Li [1] and Yong Zhang [2]

1   School of Computer Science (National Pilot Software Engineering School), University of Posts and
    Telecommunication, Beijing 100876, China
2   Zhongguancun Laboratory, Beijing 100094, China
*   Correspondence: huangxh@bupt.edu.cn

**Abstract:** In the ring-architecture-based federated learning framework, security and fairness are severely compromised when dishonest clients abort the training process after obtaining useful information. To solve the problem, we propose a **Ring-** architecture-based **F**air **F**ederated **L**earning framework called RingFFL, in which we design a penalty mechanism for FL. Before the training starts in each round, all clients that will participate in the training pay deposits in a set order and record the transactions on the blockchain to ensure that they are not tampered with. Subsequently, the clients perform the FL training process, and the correctness of the models transmitted by the clients is guaranteed by the HASH algorithm during the training process. When all clients perform honestly, each client can obtain the final model, and the number of digital currencies in each client's wallet is kept constant; otherwise, the deposits of clients who leave halfway will be compensated to the clients who perform honestly during the training process. In this way, through the penalty mechanism, all clients either obtain the final model or are compensated, thus ensuring the fairness of federated learning. The security analysis and experimental results show that RingFFL not only guarantees the accuracy and security of the federated learning model but also guarantees the fairness.

**Keywords:** federated learning; fairness; blockchain; ring architecture

## 1. Introduction

With the development of IoT and the widespread adoption of mobile devices, artificial intelligence (AI) technology is widely used in many aspects of daily life [1,2]. Considering security issues, federated learning (FL) comes into being, which enables collaborative training by exchanging local model parameters instead of raw data among devices [3,4]. The training data are kept locally during the training process, thus effectively protecting users' privacy [5,6]. This advantage in turn motivates more users to participate in FL, which leads to more accurate global models [7,8].

Traditional FL typically uses a central server to coordinate the collaboration of clients, which is called star architecture [9,10]. Specifically, in each training round, the server updates the global model by aggregating the local model parameters generated by the clients from their respective local models. The server broadcasts the updated global model to the clients. As a result, most of the clients participating in the training will make simultaneous transmissions in a short period. However, the explosive growth in traffic can have a severe impact on the network, thus limiting the number of clients that can participate in FL [11,12]. At the same time, the trustworthiness of the clients and vulnerability to single-point attacks pose hidden risks to the security and reliability of FL [13,14].

Some scholars start by changing the traditional star architecture of FL to solve the above problems and propose distributed FL architectures [15,16]. At present, distributed FL architectures are mainly divided into two categories: fully connected architectures [17–19]

and ring architectures [20–23]. Under the fully connected architecture, clients form a P2P network, in which each client transmits local model parameters to neighboring clients, receives local model parameters transmitted from neighbors to complete model aggregation, and continues to transmit until the models finally converge. However, this approach brings a large amount of redundant communication, and it is difficult to guarantee the convergence of the model because each client performs aggregation locally [24]. In the ring architecture, clients are connected in a ring communication network in a certain order, and during the training process, each client only needs to transmit the local model parameters to the next client. The next client will either continue training based on the model received or transmit to the next client after aggregating it with its local model. Through multiple rounds of loop transmission, the converged global model is finally obtained. The ring structure achieves better results due to its more efficient communication efficiency and better convergence.

In the above scenario, it is usually assumed that all clients are honest, that is, all clients participate in the training process in its entirety [25]. In fact, a client may maliciously abandon the subsequent training after obtaining a result in their favor or leave midway unintentionally for technical reasons such as unstable network conditions, and the impact of such dishonest behavior (called escape attack) on the ring structure is catastrophic [26]. Firstly, the model training may be unsustainable due to a break in the ring communication at a certain point, which will lead to low accuracy of FL. Secondly, the intermediate result obtained by the dishonest client contains data contributions from honest clients, but honest clients did not obtain any benefit, which is unfair to them. Therefore, the escape attack needs to be avoided as much as possible to ensure the fairness of all clients participating in the model training.

Motivated by the above analysis, we propose a **Ring**-architecture-based **F**air **F**ederated **L**earning framework (RingFFL), in which we design a penalty mechanism to guarantee the fairness of FL, that is, the client either obtains the model that is transmitted correctly or is compensated with a certain amount of digital currency. The HASH comparison guarantees the correctness of the models transmitted during the training process. Meanwhile, blockchain is introduced in the training process to guarantee the immutability of the clients' deposits and compensation operations. Finally, the security analysis and experimental evaluation are carried out. The main contributions of this paper are as follows:

- To ensure the fairness of FL training, we propose a penalty mechanism where clients need to pay deposits in a given order before participating in the training. If all clients perform honestly, they will all obtain the final global model, otherwise, deposits of dishonest clients will be compensated to the honest clients. In other words, clients either obtain the final global model or are compensated, thus ensuring fairness.
- To enhance the security of FL training, we propose the HASH comparison mechanism and introduce the blockchain. Through HASH comparison, the correctness of the models transmitted by the clients during the training process is guaranteed. With the immutability of the blockchain, the security of the client's deposit payment and compensation operations is guaranteed.
- To evaluate the performance of our mechanism, we perform a security analysis of the proposed mechanism and conduct experiments using MNIST and CIFAR10 data sets. The security analysis and experimental results show that our mechanism ensures security and fairness while maintaining high accuracy.

The rest of the paper is organized as follows. In Section 2, the related work is reviewed. In Section 3, the scenario, threat and adversary model, as well as the work flow of RingFFL are introduced. In Section 4, the proposed penalty mechanism is specifically described. In Section 5, the security analysis and numerical results are presented. Section 6 concludes the paper with a summary and points out the future directions.

## 2. Related Works

In this section, we briefly describe the challenges of FL first. For the challenges of poor scalability and vulnerability to single-point attacks of star architectures in FL, we provide a

detailed description of the distributed FL framework based on fully connected architecture and ring architecture.

### 2.1. Challenges of FL

Federated learning, a pioneering technique to protect data privacy, which enables training machine learning models on distributed data sets, has been widely used in various fields such as IoT, IoV, and Data Access [27]. However, FL still faces many challenges, such as expensive communication costs, systems heterogeneity, statistical heterogeneity, and privacy concerns [28]. To address the problem of expensive communication costs, Paragliola et al. proposed a new learning strategy that reduced the total number of parameters shared in the FL process and evaluated the trade-off between the requirement to reduce communication costs and the need to guarantee the highest classification performance [29]. To address the problem of system heterogeneity, Lin et al. proposed an ensemble refinement scheme for model aggregation, where a central classifier was trained from unlabeled data of the clients' model output, a technique that allowed flexible aggregation of heterogeneous client models [30]. To address the statistical heterogeneity problem, Liu et al. proposed a client–edge–cloud hierarchical FL system supported by the HierFAVG algorithm for better communication–computation tradeoffs [31]. To address the problem of data heterogeneity, Paragliola et al. investigated and evaluated the performance and behavior of a federated model in the presence of catastrophic forgetting events in the context of nonstationary data [32]. Huang et al. illustrated how data skewing could affect the performance of FL algorithms and then proposed a new algorithm, FedMix, which adapted existing FL algorithms and achieved better performance [33]. For security issues, Li et al. proposed q-Fair Federated Learning (q-FFL) for fairer accuracy allocation among devices, which outperformed the existing baseline in terms of fairness, flexibility, and efficiency [34]. Fang et al. proposed a multiparty privacy-preserving machine learning framework (PFMLP) based on partial homomorphic encryption and FL to achieve protection of data and model parameters during training [35].

### 2.2. Distributed FL Framework

The distributed FL framework based on fully connected architecture completes model training through P2P model transmission between clients, and the implementation of the architecture often requires the support of blockchain, Device-to-Device (D2D), and so on [17–19]. Samarakoon et al. proposed a novel method based on distributed FL to estimate the tail distribution of vehicle formation lengths, approaching the accuracy of the centralized solution while reducing the amount of data exchanged by 79% [17]. Li et al. proposed a decentralized blockchain-based FL framework called the Blockchain-based Federated Learning framework with Committee consensus (BFLC), which avoided attacks by malicious central servers on global models and users' private data, and designed an innovative committee consensus mechanism for BFLC that could effectively reduce consensus computation and malicious attacks [18]. Qu et al. proposed a novel blockchain-based FL scheme called FL-Block that balanced the privacy protection of fog computing with the attendant inefficiencies, excelling in privacy protection, efficiency, and resistance to poisoning attacks [36]. Xing et al. proposed distributed stochastic gradient descent (DSGD) to achieve large-scale deployment of FL in wireless communication scenarios [37]. Zhang et al. proposed an FL scheme using D2D communication (D2D-FedAvg), where D2D grouping, primary UE selection, and D2D exit were implemented in the learning process, resulting in a complete D2D-assisted FL averaging algorithm [19]. Although the problem of poor scalability and vulnerability to single-point attacks of star architectures is solved to some extent by using fully connected architecture, it brings a lot of redundant communication consumption, and it is difficult to guarantee the convergence of the model due to the lack of global aggregation.

Some scholars have designed a ring architecture to complete the training through the ring transmission of models while ensuring lower communication redundancy and higher

convergence [20–23]. Xu et al. proposed an optimization method to minimize FL uplink transmission time in wireless networks and designed a ring all-reduce architecture based on D2D, which significantly reduced the transmission time [20]. Wang et al. proposed a ring-topology-based distributed FL (RDFL) scheme for deep generative modeling (DGM), which provided communication efficiency and maintained training performance to boost DGMs in target tasks compared with existing FL works [21]. Lee et al. designed a cyclic learning scheme based on the ring topology to reduce the number of FL training iterations and improve the performance without adding any additional computational cost [22]. Lee et al. proposed a new algorithm called TornadoAggregate to improve FL accuracy and scalability by promoting a ring architecture. Experimental results show that TornadoAggregate improved test accuracy by 26.7% and achieved near-linear scalability [23].

The above schemes usually assume that the clients are honest. However, the assumption is difficult to be satisfied in realistic situations. A malicious client sending escape attacks during the training process can be fatal to FL based on the ring architecture. Therefore, we propose a fair FL framework to solve the problem. With the proposed penalties mechanism, clients are effectively prevented from escaping from the training after obtaining valid information.

## 3. Framework Overview

In this section, we focus on the scenario, threat, and adversary model and workflow of the proposed RingFFL.

### 3.1. The Scenario of RingFFL

We consider the scenario where a ring architecture is formed among different clients to perform FL and no server exists. Without loss of generality, we assume that there are $N$ clients. All clients involved in FL first agree on the order of the ring architecture. As shown in Figure 1, the black dashed line indicates the communication link between clients. The whole training process is divided into two phases, in which one is the deposit payment phase (red dashed line) and the other is the model transmission phase (blue solid line). In the deposit payment phase, all clients will prepay the deposits according to the proposed mechanism. In the model transmission phase, the local model parameters of the previous client are sent to the next client. After all clients obtain the global model for one round of FL, they will continue training according to the same rules until a predefined number of rounds or a loss function convergence threshold is reached.

### 3.2. Threat and Adversary Model

**Threat Model:** In our mechanism, there are $N$ clients cooperating to train the FL model. We assume that all clients train local models honestly, and a subset of them are malicious or are potentially corrupted by a malicious adversary when transmitting local model parameters.

**Adversary Goal:** The adversary has two goals. One is to manipulate clients to send false local model parameters to mislead the global model of FL. The other is to abort during training to obtain useful intermediate information without effort, which can lead to some clients' contributing but no benefit, and some clients cannot obtain the FL model. Both threats will not only reduce the accuracy of FL but also destroy the fairness.

### 3.3. The Work Flow of RingFFL

The work flow of RingFFL is shown in Figure 2. Suppose there are $N$ distributed clients participating in the FL. In order to ensure the fairness, RingFFL is designed based on the blockchain. It is divided into four steps.

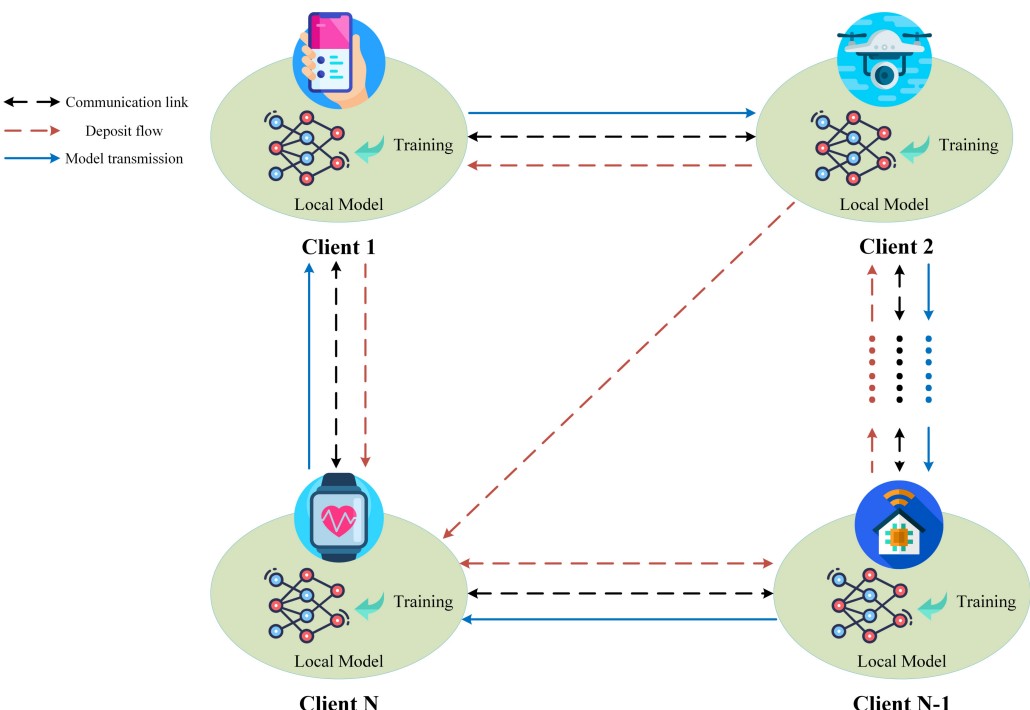

**Figure 1.** The scenario of RingFFL.

**Step 1**   *Initialization*

Each client generates a random number to form a private key, computes the corresponding public key, and produces a digital currency wallet address. Apart from this, they perform data preprocessing on their own data sets.

**Step 2**   *Local Training*

Each client trains a local model based on its private data set. Since all clients do not want to reveal their sensitive information, we use FL to protect the clients' raw data. The local model parameters sent by the clients during the FL process can be protected using many methods, such as differential privacy [13]. It is not the focus of our approach and is not described in detail.

**Step 3**   *Transactions on the blockchain*

Transactions on the blockchain phase are divided into three parts: roof deposits, ladder deposits, and acknowledgment. We introduce them below.

**(3.1)**   *Roof deposits*

According to the topology of the ring structure, all clients agree on a sequence. Except for the last client, all the others make the same amount of deposits to it and broadcast all transactions to the network.

**(3.2)**   *Ladder deposits*

Each client pays some deposits according to the predetermined rule to its previous client. All transactions will be broadcasted to the network.

**(3.3)**   *Acknowledgement*

All clients will provide their local model parameters to acknowledge digital currencies deposited in (3.1) and (3.2). Successful transactions will be recorded on the blockchain.

**Step 4**    *Output*

All local model parameters will be aggregated to generate the global model in a round and shared among all clients. The process will be repeated until a predetermined number of training rounds or a convergence threshold is reached. If all clients behave honestly, each client's digital currencies will neither increase nor decrease in the end. Otherwise, dishonest clients' deposits will be used to compensate for every honest client that does not obtain the final model of the FL.

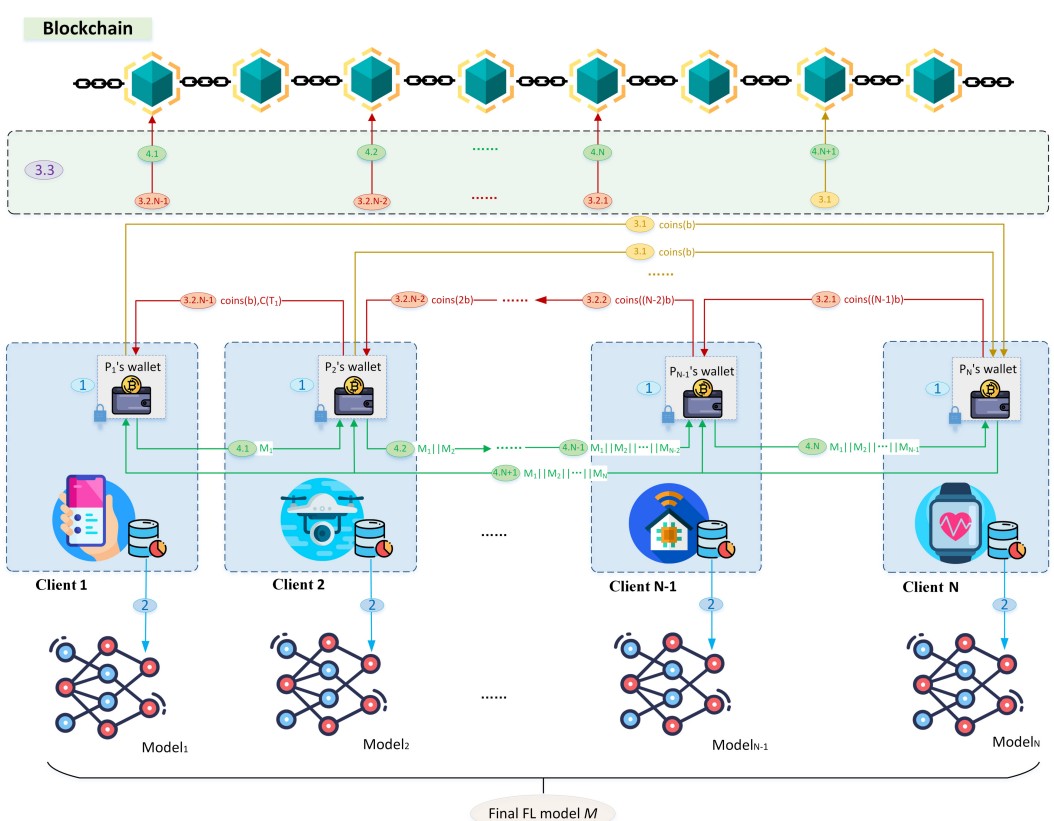

**Figure 2.** The work flow of RingFFL.

## 4. Penalty Mechanism in RingFFL

In FL, each distributed client trains a local model according to its data set locally. Then, all the local model parameters can be safely aggregated into a global model to be used for more accurate prediction. However, each client may be corrupted and does not share their local model with other clients. In this case, they can synthesize a more accurate model to improve their prediction accuracy, while other clients cannot, although they share their local models.

To deal with this, we put forward RingFFL to ensure that if a dishonest client refuses to send their local model to honest clients after receiving others' local models, the honest ones who do not obtain the results will obtain a certain amount of digital currency compensation provided by the dishonest ones. The proposed mechanism is applicable to FL with ring architecture, and clients that can form the ring topology can participate in FL, such as mobile devices, IoT terminals, and self-driving vehicles. In the paper, we deal only with the static adversaries and assume that all clients will honestly train their local models. The main notations used are shown in Table 1.

**Table 1.** Summary of main notations.

| Parameters | Description |
|:---:|:---:|
| $P_i$ | The $i$-th client |
| $N$ | The number of all clients |
| $T_i$ | Client $i$'s private data set |
| $M_i$ | Client $i$'s local model parameters |
| $C$ | Conditions for obtaining digital currencies |
| $E$ | Evidence needed to obtain digital currencies |
| $M^{int}$ | Initial model of FL |
| $MAX$ | Maximum communication rounds of FL |

To gain a better understanding of the mechanism, motivated by [38,39], we introduce the process of digital currency changing on the blockchain. Firstly, sender $S$ of the deposits broadcasts a transaction to the network. The transaction involves the number of deposits $coins(b)$, a boolean circuit $C(\cdot)$, and the time limit $t$. Secondly, the recipient $R$ of the deposits provides evidence $e$ in time $t$. Finally, if $e$ satisfies $C(e) = 1$, the transaction will be recorded on the blockchain, and $coins(b)$ will be sent to $R$'s wallet. If $R$ cannot provide $e$ or $C(e) = 0$ in time $t$, then the transaction will not be recorded, and $coins(b)$ will be returned to $S$'s wallet. The penalty mechanism is detailed below.

**Step 1** *Initialization*

At first, each client uses the SHA256 algorithm [40] to generate a 256-bit random number as the private key, which is used to generate the corresponding public key and used for digital signatures in the transactions. Next, they calculate the corresponding public key according to the Elliptic Curve Cryptography algorithm [41]. Once more, based on the RIPEMD160 [42] and SHA256 algorithms [40], each client generates a digital currency wallet address. Each client must use their private key for signature authentication to transmit the digital currencies in their wallet. At last, each client performs data preprocessing on their own data set, which mainly includes data cleaning and data integration.

**Step 2** *Local training*

The data sets of the clients $P_i$ are $T_i (i = 1, 2, \cdots, N)$, respectively. In the local training phase, all $P_i (i = 1, 2, \cdots, N)$ construct local model parameters $M_i (i = 1, 2, \cdots, N)$ according to $T_i$. Then, each client calculates

$$H_i = Hash(M_i) \tag{1}$$

and outputs $\{H_i\}_{i \in [N]}$ to $P_i (i = 1, 2, \cdots, N)$.

**Step 3** *Transactions on the blockchain*

In our mechanism, paying deposits is a necessary condition for clients to participate in the FL, even for honest clients. It is believed that the more digital currency clients hold, the more reliable they are. Clients who leave frequently midway will hold less and less digital currency and cannot even meet the deposit amount required to participate in the FL, and thus cannot participate in the training. Transactions on the blockchain phase consist of three parts, which are introduced in detail below.

**(3.1)** *Roof deposits*

We introduce the roof deposits phase in two cases: normal and abnormal situations. The normal situation is the case that all clients perform honestly as required, and the abnormal situation is the case that there are clients who do not perform the corresponding process as required during the training. The situations are presented in detail below.

(a) Normal situation.

Each client $P_i (i = 1, \cdots, N - 1)$ simultaneously deposits $coins(b)$ to $P_N$ under the condition $C_N$, and the deadline is $t_N$. The detail is that each $P_i (i = 1, 2, \cdots, N - 1)$ broadcasts

a transaction $\left(deposit, T_{id}, T'_{id}, P_i, P_N, C_N, t_N, b\right)$ to the network, in which $T_{id}$ is the session identifier and $T'_{id}$ is the transaction identifier. Ignore other transactions with the same $T'_{id}$ sent by $P_i (i = 1, 2, \cdots, N-1)$ to $P_N$ before the end of RingFFL. The process is denoted as

$$Ack_{i,N}: \quad P_i \xrightarrow[b,t_N]{C_N} P_N \tag{2}$$

(b)  Abnormal situation.

If the transaction $\left(deposit, T_{id}, T'_{id}, P_i, P_N, C_N, t_N, b\right)$ is not recorded on the blockchain for any one of $P_i (i = 1, \cdots, N-1)$, all $P_i (i = 1, \cdots, N-1)$ wait until time $t_{N+1}$ to receive $Ack_{i,N}$'s reimbursement message $\left(reimburse, T_{id}, T'_{id}, P_i, P_N, C_N, t_N, coins(b)\right)$ to $P_i$'s wallet, and the RingFFL stops, in which $i = 1, \cdots, N-1$.

**(3.2)**  *Ladder deposits*

Like roof deposits, we introduce the ladder deposits phase in two cases: normal and abnormal situations.

(a)  Normal situation.

After the clients, except $P_N$, complete the roof deposits, all clients perform the process of ladder deposits in turn. First, $P_N$ broadcasts a transaction $\left(deposit, T_{id}, T'_{id}, P_N, P_{N-1}, C_{N-1},\right.$ $\left. t_{N-1}, (N-1)b\right)$ to the network as a deposit to $P_{N-1}$ under condition $C_{N-1}$ with deadline $t_{N-1}$. Similarly, $P_{N-1}$ deposits $coins((N-2)b)$ to $P_{N-2}$ under condition $C_{N-2}$. It continues until $P_2$ deposits $coins(b)$ to $P_1$ under condition $C_1$. For $i = N-1$ down to $i = 1$, the process is denoted as

$$Ack_{i+1,i}: \quad P_{i+1} \xrightarrow[i \cdot b, t_i]{C_i} P_i \tag{3}$$

(b)  Abnormal situation.

If there is no transaction within the network that client $P_{i+1}$ sends deposits to $P_i$ $(i = 1, \cdots, N-1)$ as described above, all $P_j (j \leq i)$ do not perform the ladder deposits and wait for $Ack_{i,N}$'s reimbursement according to Equation (2). All $P_j (j > i)$ need to wait until the end of the RingFFL to judge whether their ladder deposits are acknowledged.

**(3.3)**  *Acknowledgment*

Like roof deposits, we introduce the acknowledgment phase in two cases: normal and abnormal situations.

(a)  Normal situation.

In the acknowledgment phase, we define $E_{i+1}$ and $C_i$ as follows:

$$E_{i+1} \triangleq Expand(i+1, E_i; M_{i+1}) = E_i \| M_{i+1} \tag{4}$$

$$C_i\left(M'_1 \| \cdots \| M'_i; \{H_1, H_2, \cdots, H_i\}\right) = \left[Hash\left(M'_1\right) \underline{?} H_1\right] \cap \cdots \cap \left[Hash\left(M'_i\right) \underline{?} H_i\right] \tag{5}$$

in which $H_i$ is associated with Equation (1), $M'_i$ is the model parameters received from neighboring client, and $A \| B$ denotes a connection between $A$ and $B$.

$P_1$ first provides evidence $E_1 = Expand(1, \perp; M_1)$ according to Equation (4) to acknowledge $Ack_{2,1}$ from $P_2$. When $P_1$ provides the acknowledgment message $\left(acknowledge, T_{id}, T'_{id}, P_2, P_1, C_1, t_1, b, E_1\right)$, miners will verify whether the transaction $\left(deposit, T_{id}, T'_{id}, P_2, P_1, C_1, t_1, b\right)$ exists and that $C_1(E_1; H_1) = 1$ is true according to Equa-

tion (5). If the verification passes, the message $\left(acknowledge, T_{id}, T'_{id}, P_2, P_1, C_1, t_1, b, E_1\right)$ is made public to the network, and the transaction $\left(deposit, T_{id}, T'_{id}, P_2, P_1, C_1, t_1, b\right)$ is recorded on the blockchain later. The $coins(b)$ in $\left(acknowledge, T_{id}, T'_{id}, P_2, P_1, C_1, t_1, coins(b)\right)$ is sent to $P_1$'s wallet. Afterwards, $P_2$ acknowledges $Ack_{3,2}$ from $P_3$, similarly. It continues in turn until $P_{N-1}$ acknowledges $Ack_{N,N-1}$ from $P_N$ using evidence $E_{N-1}$ according to Equations (4) and (5). At last, $P_N$ uses $E_N$ to acknowledge $Ack_{i,N}$ (Equation (2)) from all $P_i(i = 1, 2, \cdots, N-1)$.

(b)  Abnormal situation.

For $i = 1, 2, \cdots, N-1$, if $P_i$ does not acknowledge $Ack_{i+1,i}$ according to Equation (3), then $P_{i+1}$ will end the RingFFL and wait for reimbursement from $Ack_{i+1,i}$. Furthermore, when $i + 1 \neq N$, $P_{i+1}$ waits for reimbursements from $Ack_{i+1,N}$ according to Equation (2).

**Step 4**  *Output*

Like roof deposits, we introduce the output phase in two cases: normal and abnormal situations.

(a)  Normal situation.

In the output phase, if $P_N$ acknowledges $Ack_{i,N}$ according to Equation (2)), then $E_N$ (Equation (4)) is made public, which is the evidence that meets $C_N$(Equation (5)). Each $P_i(i = 1, \cdots, N)$ outputs $M_1\|\cdots\|M_N$, uses the FedAvg algorithm [43] for model aggregation to obtain the current round of global model $M_{Round}^{glob} = \frac{1}{N}\sum_N M_i$, and continues to the next round of training.

In the RingFFL, for $i = 1, \cdots, N-1$, $Ack_{i,N}$ (Equation (2)) locks $coins(b)$ of $P_i$. If $P_{i-1}$ successfully executes $Ack_{i,i-1}$'s acknowledgment (Equation (3)), then $P_i$ will give $P_{i-1}$ $coins((i-1)b)$ and the condition $C_{i-1}$ (Equation (5)) provided by $P_{i-1}$ will be made public. According to the model parameters $M_i$ of $P_i$ and $C_{i-1}$, $P_i$ can successfully execute $Ack_{i+1,i}$'s acknowledgment (Equation (3)) and obtain $coins(ib)$ from $P_{i+1}$. That is, if all clients execute the RingFFL honestly, the number of each client's digital currencies neither increase nor decrease.

(b)  Abnormal situation.

If $P_N$ does not acknowledge any $Ack_{i,N}$ (Equation (2)), then each client outputs $\perp$.

For $i = 1, \cdots, N$, suppose $P_j(j \in [i-1])$ are honest clients. If $P_i$ terminates the RingFFL after obtaining $M_1\|M_2\|\cdots\|M_{i-1}$, then $P_{i-1}$ can successfully acknowledge $coins((i-1)b)$ from $P_i$'s ladder deposits, and $P_k$'s $coins((k-1)b)$ in the ladder deposits will be reimbursed, in which $k = i+1, i+2, \cdots, N$. Since $P_N$ cannot provide $C_N$ (Equation (5)), each client's roof deposits will be reimbursed. As a result, the total number of $P_k$'s digital currencies is unchanged, in which $k = i+1, i+2, \cdots, N$. The number of each $P_j$'s digital currencies increases by $b$, and the number of $P_i$'s digital currencies decreases by $(i-1)b$. Thus, the RingFFL guarantees the fairness of FL. The overall process of RingFFL is illustrated in Algorithm 1.

To better understand the whole process, we take 5 clients as an example to demonstrate the penalty mechanism in RingFFL, which is shown in Figure 3.

---

**Algorithm 1** The proposed RingFFL

---

**Input:** $\{T_1, T_2, \cdots, T_N\}$, $M^{int}$, $MAX$.
**Output:** Federated Learning Model $M$.
1: Set $Round = 1$
2: **for** $i = 1 : N$ **do**
3: $\quad$ $P_i$ trains local models $M_i$ based on initial model $M^{int}$, calculates $H_i$, and broadcasts
$\quad\quad$ $\{H_i\}_{i \in [N]}$ to $P_i$.
4: **end for**
5: **for** $i = 1 : N - 1$ **do**
6: $\quad$ $P_i$ deposits $coins(b)$ to $P_N$ under the condition $C_N$, which is denoted as $Ack_{i,N}$.
7: $\quad$ **if** $P_i$ is honest **then**
8: $\quad\quad$ Process $Ack_{i,N}$ is executed.
9: $\quad$ **else**
10: $\quad\quad$ Transaction $\left(deposit, T_{id}, T'_{id}, P_i, P_N, C_N, t_N, b\right)$ is not recorded on the blockchain.
11: $\quad\quad$ **for** $i = 1 : N - 1$ **do**
12: $\quad\quad\quad$ $P_i$ waits to receive $Ack_{i,N}$'s reimbursement message $\left(reimburse, T_{id}, T'_{id}, P_i,\right.$
$\quad\quad\quad\quad$ $\left. P_N, C_N, t_N, coins(b)\right)$.
13: $\quad\quad\quad$ End RingFFL
14: $\quad\quad$ **end for**
15: $\quad$ **end if**
16: **end for**
17: **for** $i = 1 : N - 1$ **do**
18: $\quad$ $P_{i+1}$ deposits $coins(i \cdot b)$ to $P_i$ under the condition $C_i$, which is denoted as $Ack_{i+1,i}$.
19: $\quad$ **if** $P_i$ is honest **then**
20: $\quad\quad$ Process $Ack_{i+1,i}$ is executed.
21: $\quad$ **else**
22: $\quad\quad$ **for** $j = 1 : i - 1$ **do**
23: $\quad\quad\quad$ $P_j$ does not perform the ladder deposits and wait for $Ack_{i,N}$'s reimbursement.
24: $\quad\quad$ **end for**
25: $\quad\quad$ **for** $j = i + 1 : N - 1$ **do**
26: $\quad\quad\quad$ $P_j$ wait for the end of RingFFL to judge whether its ladder deposit is acknowl-
$\quad\quad\quad\quad$ edged.
27: $\quad\quad$ **end for**
28: $\quad$ **end if**
29: **end for**
30: **for** $i = 1 : N$ **do**
31: $\quad$ **if** $P_i$ is honest **then**
32: $\quad\quad$ $P_i$ outputs $M^{glob}_{Round} = \frac{1}{N} \sum_N M_i$
33: $\quad$ **else**
34: $\quad\quad$ $P_i$ outputs $\perp$
35: $\quad\quad$ End RingFFL
36: $\quad$ **end if**
37: **end for**
38: **if** Round<MAX **then**
39: $\quad$ $Round = Round + 1$
40: $\quad$ Return 2
41: **else**
42: $\quad$ **for** $i = 1 : N$ **do**
43: $\quad\quad$ $P_i$ outputs $M = M^{glob}_{Round}$
44: $\quad\quad$ End RingFFL
45: $\quad$ **end for**
46: **end if**

---

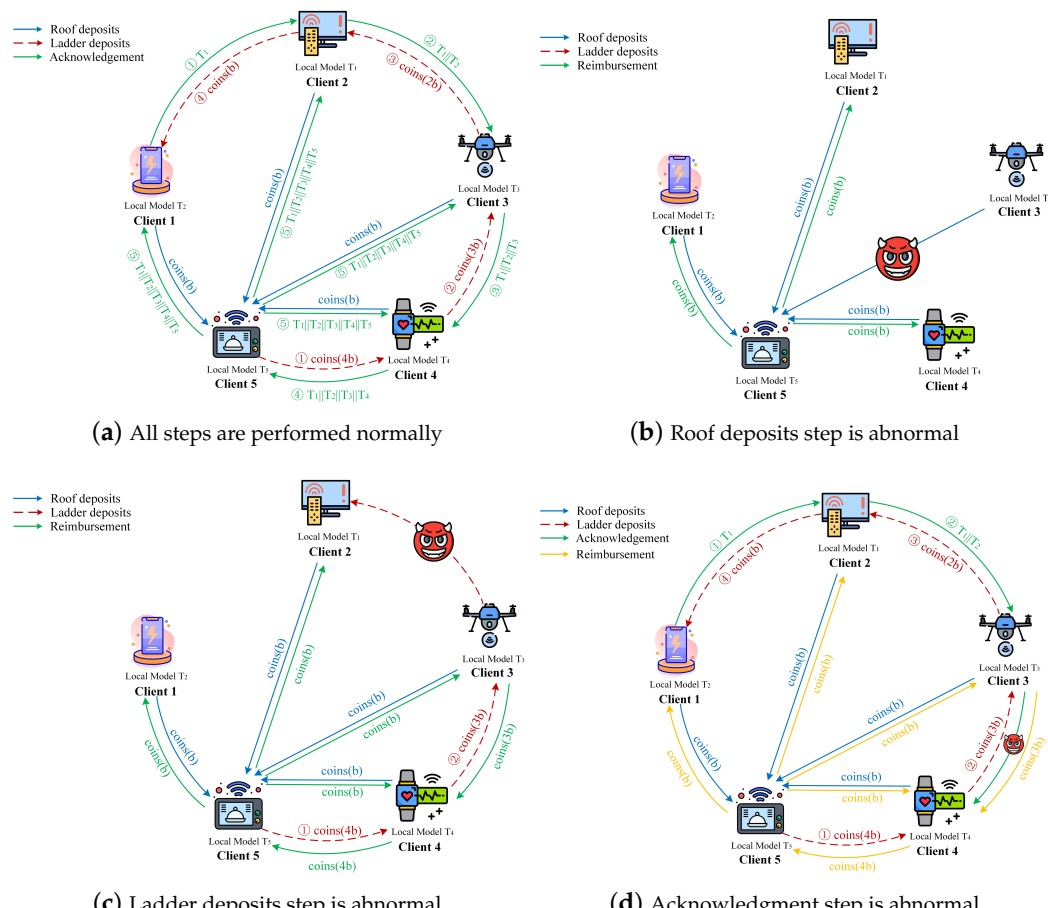

(**a**) All steps are performed normally

(**b**) Roof deposits step is abnormal

(**c**) Ladder deposits step is abnormal

(**d**) Acknowledgment step is abnormal

**Figure 3.** An example of the penalty mechanism in RingFFL.

## 5. Security Analysis and Numerical Results

In the section, we focus on the security analysis and numerical results of the proposed RingFFL.

### 5.1. Security Analysis

In this paper, we use blockchain in the ring-architecture-based FL to address the aforementioned security threats.

(1) *Guaranteeing the data security of the clients*: We use FL with ring architecture to train the model, and the local model parameters instead of the raw data are transmitted during the training process. The raw data are stored locally from start to finish, thus ensuring the security of the clients' data.

(2) *Guaranteeing the security of the clients' deposits*: In the process of deposit payment and deposit refund, we use blockchain to guarantee the security of the transactions. The security mechanism based on consensus can avoid security attacks such as double spending, thus ensuring the security of clients' deposits during the training process.

(3) *Guaranteeing the correctness of the transmitted local model parameters*: To prevent the dishonest clients from transmitting false local model parameters, the acknowledgment process needs to validate the hash values of the model parameters, and only the correct ones will pass the validation. The client will then receive the corresponding amount of digital currencies.

(4) *Guaranteeing the fairness of the FL*: To avoid dishonest clients from aborting during the FL process, the penalty mechanism can deduct the digital currency deposits of the aborting clients to compensate clients with additional information loss. In this way, the fairness of FL is guaranteed.

### 5.2. Numerical Results

We perform experiments using two well-known data sets MNIST [44] and CIFAR10 [45] to verify the effect of RingFFL, which are widely used in FL. MNIST is a handwritten digit data set, containing 60,000 training samples and 10,000 test samples, in which each image is a single digit from 0 to 9. The CIFAR10 data set consists of 60,000 color images of 10 classes, which contains 50,000 training images and 10,000 test images and each image is $32 \times 32$ in size. According to the distribution of data among clients, we conducted experiments for the cases where the data sets of clients are independent and identically distributed (IID) [46] and nonindependent and identically distributed (non-IID) [47], respectively. Whether clients behave honestly or dishonestly in FL is independent of the distribution of their data sets. When simulating the case of IID, we uniformly sample the data with labels 0–9 as the data set for each client, and the distribution of clients' training data obeys a uniform distribution. In the case of simulating the situation of non-IID, we first preprocess the data by dividing the original data into 20 parts, in which each containing half of all the data in a particular category. Then, we put every 2 parts together as a client's data set according to $(i, j)$ categories, where $i \neq j$, which is a common method to simulate non-IID in FL.

To demonstrate the accuracy of RingFFL for MNIST and CIFAR10 data sets in IID and non-IID cases, respectively, Figures 4 and 5 are given.

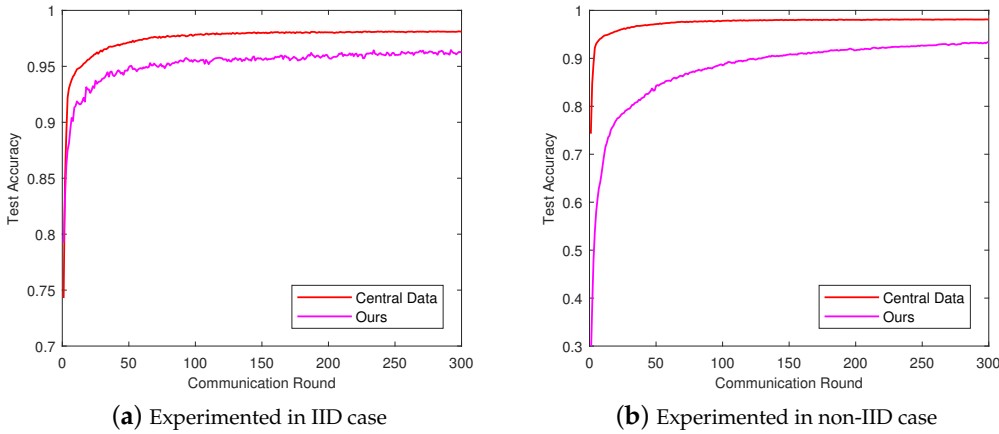

| (a) Experimented in IID case | (b) Experimented in non-IID case |

**Figure 4.** Accuracy comparison of FL for MNIST data set in IID and non-IID cases.

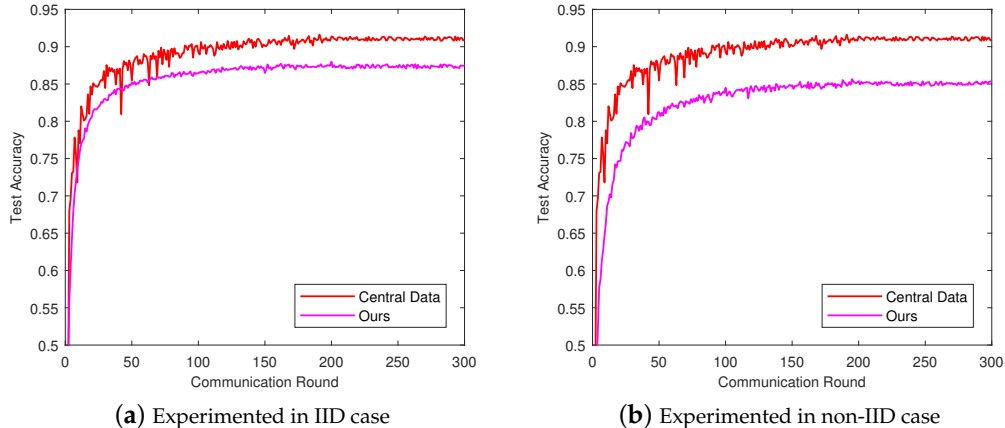

| (a) Experimented in IID case | (b) Experimented in non-IID case |

**Figure 5.** Accuracy comparison of FL for CIFAR10 data set in IID and non-IID cases.

Figures 4 and 5 compare the accuracy with the communication round for the centralized algorithm (Central Data in the figures) and our approach in the case of IID and non-IID experimented with MNIST and CIFAR10 data set, respectively. It can be seen that the accuracy of RingFFL gradually increases as the number of communication rounds increase.

After reaching a certain number of communication rounds, the accuracy tends to be stable and close to the accuracy of the centralized algorithm. Since CIFAR10's images are derived from the real world, the scale, features, and colors of objects are different in addition to having a lot of noise, which greatly increases the difficulty of recognition. Nevertheless, the accuracy of the experiments conducted by CIFAR10 is close to the accuracy of the centralized algorithm both in the IID and non-IID cases, which meets the requirements of FL.

To demonstrate the effect of escape attacks on the FL, Figures 6 and 7 are given. In our experiments, we assume that there are clients left in the 5th communication round of FL.

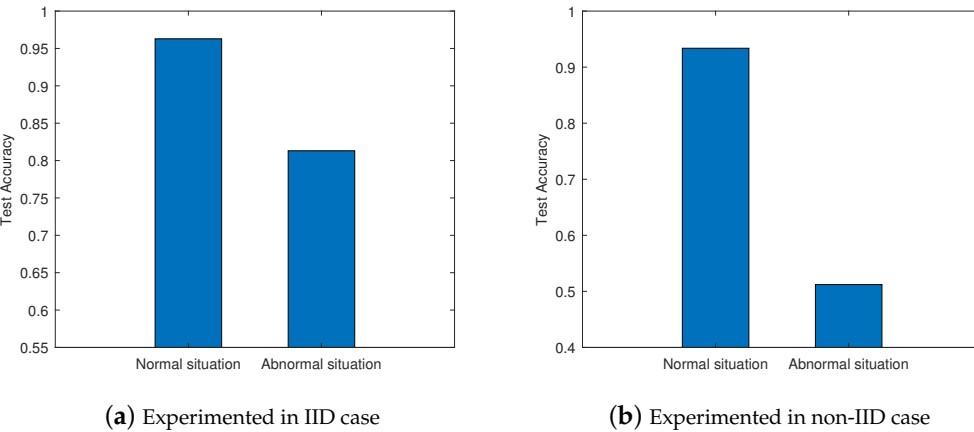

(**a**) Experimented in IID case      (**b**) Experimented in non-IID case

**Figure 6.** Accuracy comparison of FL for MNIST data set in IID and non-IID cases with clients leaving.

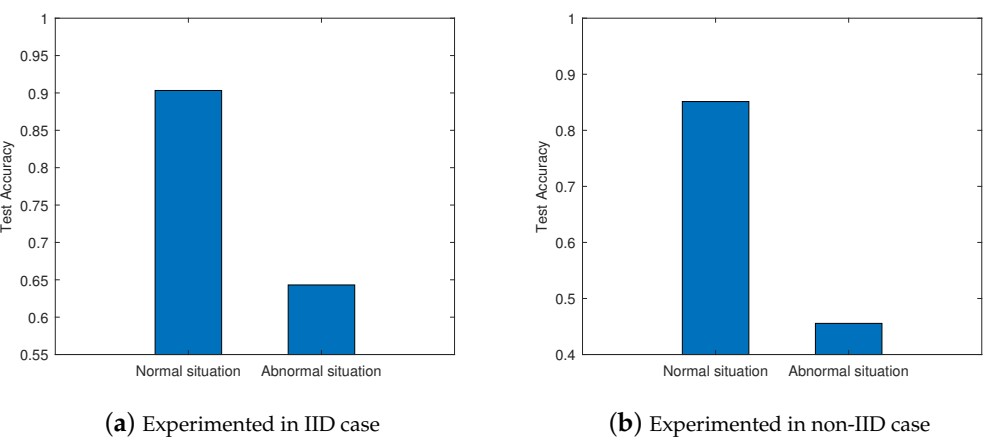

(**a**) Experimented in IID case      (**b**) Experimented in non-IID case

**Figure 7.** Accuracy comparison of FL for CIFAR10 data set in IID and non-IID cases with clients leaving.

Figures 6 and 7 compare the accuracy of the experiments using the MNIST and CIFAR10 data sets in normal and abnormal (with clients leaving) situations in IID and non-IID cases, respectively. It can be seen that the accuracy of the experiments using the MNIST and CIFAR10 data sets decreases substantially when there are clients leaving during the FL process, both in the IID and non-IID cases. For example, in the IID case of the experiment using the MNIST data set, the accuracy of FL in the normal situation is 96.29%, while the accuracy of FL in the abnormal situation with clients leaving is 81.31%. Similarly, in the non-IID case of the experiment using the CIFAR10 data set, the accuracy of FL in the normal situation is 85.13%, while the accuracy of FL in the abnormal situation with clients leaving is 45.56%.

To demonstrate the fairness of the RingFFL, we performed simulation experiments. Suppose there are 5 clients participating in the FL. The situation where all clients are honest is shown in Figure 8a. The situations where a client leaves during the roof deposit and

ladder deposit are shown in Figure 8b,c, respectively. The situation where a client leaves after obtaining useful information is shown in Figure 8d.

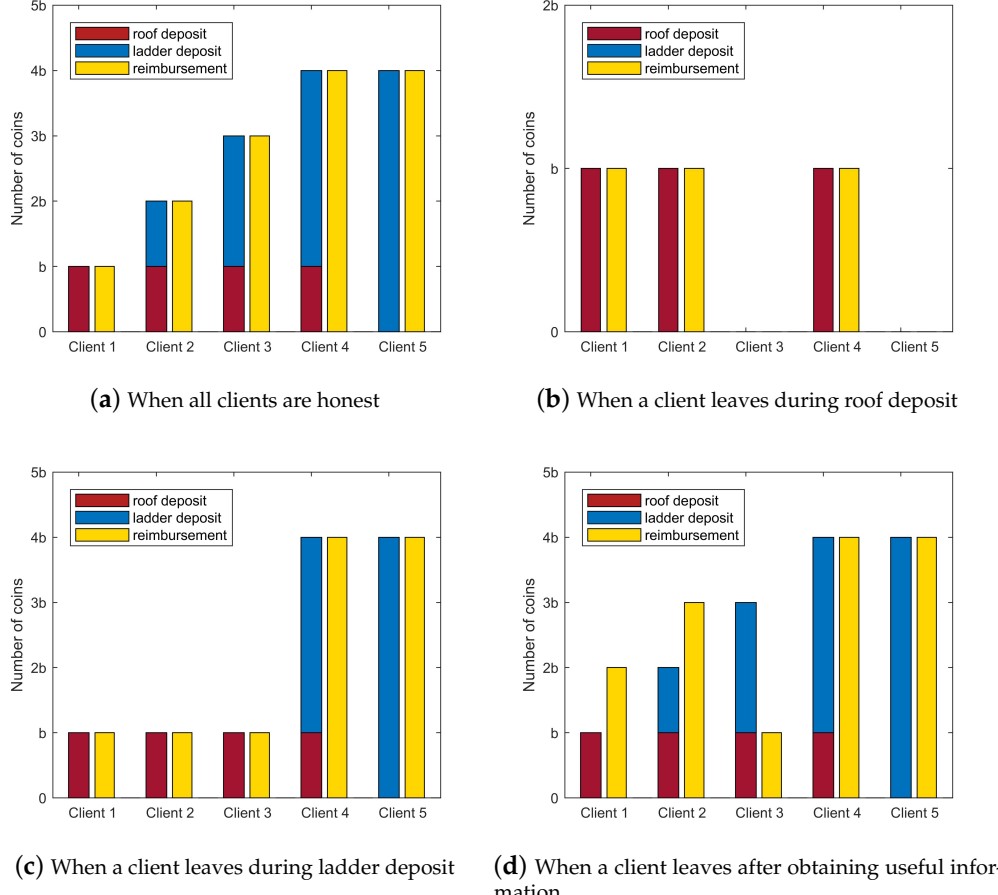

(**a**) When all clients are honest

(**b**) When a client leaves during roof deposit

(**c**) When a client leaves during ladder deposit

(**d**) When a client leaves after obtaining useful information

**Figure 8.** The circulation of digital currencies when there are 5 clients participating in FL.

As shown in Figure 8a, when all clients behave honestly, the number of digital currencies in each client's wallet neither increases nor decreases at the end of FL. Client 1's reimbursement comes from client 2's ladder deposits of $b$, and similarly, client 2, client 3, and client 4's reimbursement come from client 3, client 4, and client 5's ladder deposits of $2b$, $3b$ and $4b$, respectively. Client 5's reimbursement comes from client 1–client 4's roof deposits of $b \times 4 = 4b$.

The situation where a client aborts during the roof deposit is shown in Figure 8b. Suppose that client 3 leaves. After client 3 aborts during the roof deposit, RingFFL stops and does not continue the following process. Since no local model parameters are transmitted, client 1, client 2, and client 4's roof deposits of $b$ will be fully refunded.

The situation where a client aborts during the ladder deposit is shown in Figure 8c. Suppose that client 3 leaves. As can be seen from Figure 8c, after client 3 aborts during the ladder deposit, client 1 and client 2 will also not start the ladder deposit phase. Because no local model parameters are transmitted, ladder deposits for client 4 of $3b$ and client 5 of $4b$ and roof deposits for client 1–client 4 of $b$ will be refunded.

The situation where a client aborts during the ladder deposit is shown in Figure 8d. Suppose that client 3 leaves after obtaining useful information. It can be seen from Figure 8d that, in the case where client 3 aborts after obtaining useful information, since client 3 does not transmit the model parameters to client 4, they do not obtain the number of ladder deposits of $3b$ from client 4, which will be returned to client 4. Similarly, the number of ladder deposits of $4b$ from client 5 to client 4 will also be returned to client 5. Client 1–client 4's roof deposits to client 5 will also be returned in full. However, client 1 correctly

transmitted the model parameters to client 2, so client 1 received both client 2's ladder deposits of $b$ and client 5's returned roof deposits of $b$. That is, the number of digital currencies of client 1 eventually increases by $b$. Similarly, client 2 correctly transmits the model parameters to client 3, so client 2 obtains both the ladder deposits of $2b$ from client 3 and the roof deposits of $b$ refunded from client 5. After subtracting the number of ladder deposits of $b$ given by client 2 to client 1, the number of digital currencies in client 2's wallet also increases by $b$ in the end. In addition, as can be seen in Figure 8d, client 3's digital currencies are reduced by $2b$, while the digital currencies of client 1 and client 2, who trained honestly and transmitted the local model parameters to the next client, are both increased by $b$. This corresponds to client 3's deposits being used to compensate for the losses of client 1 and client 3. Client 4 and client 5 receive neither the model parameters nor the additional digital currencies. Thus, the fairness of FL is achieved.

To illustrate the universality of the proposed mechanism to the number of clients, we increased the number of clients and reconduct the experiments. As is shown in Figures 9–11, similar results are obtained with 10 clients, 15 clients, and 20 clients participating in FL training, respectively, in which we assume that client 8 leaves halfway.

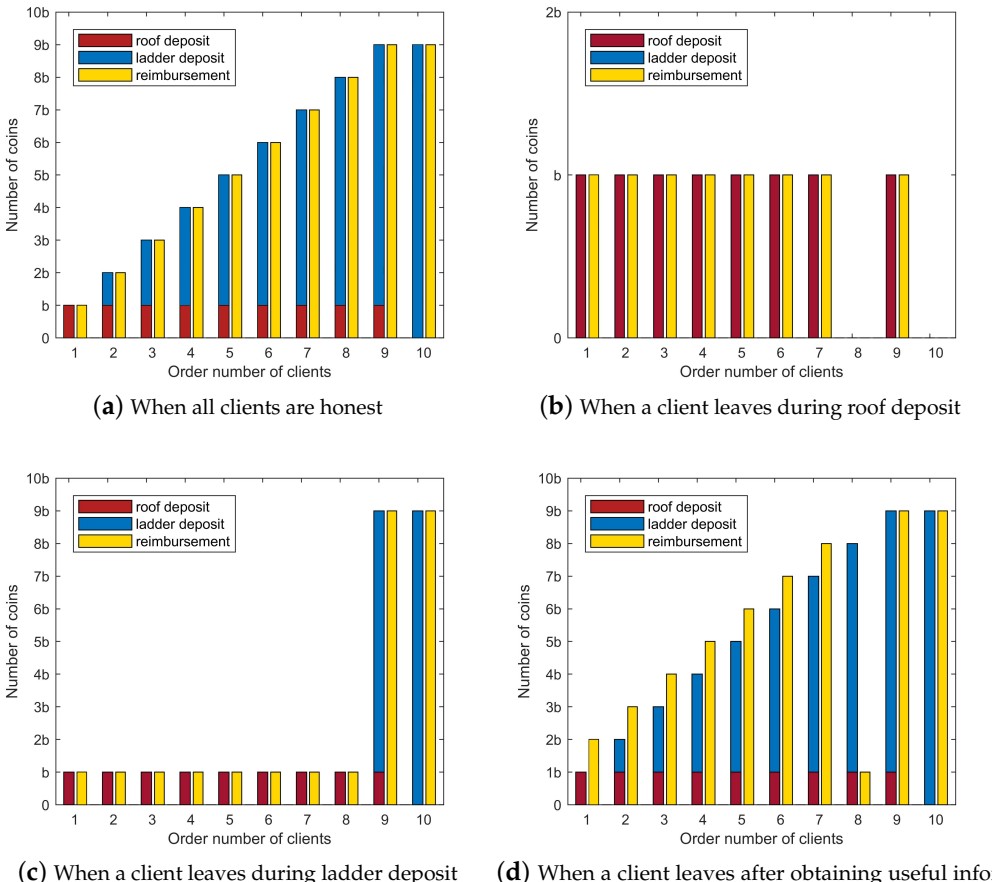

(**a**) When all clients are honest

(**b**) When a client leaves during roof deposit

(**c**) When a client leaves during ladder deposit

(**d**) When a client leaves after obtaining useful information

**Figure 9.** The circulation of digital currencies when there are 10 clients participating in FL.

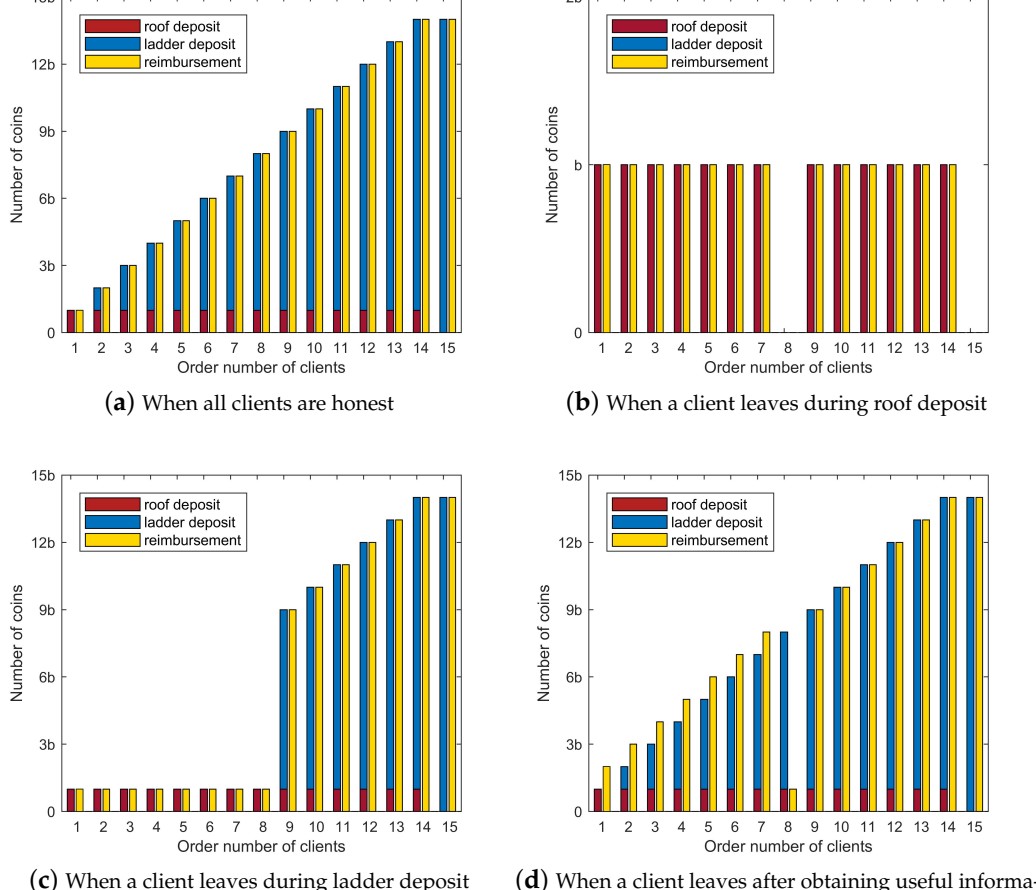

**Figure 10.** The circulation of digital currencies when there are 15 clients participating in FL.

Figures 8–11 show the case where one client leaves halfway. When more than one client leaves, the penalty mechanism still works. We conducted a simulation experiment with three clients leaving midway (client 8, client 14, and client 18), and the experimental results are shown in Figure 12.

It can be seen from Figure 12a that when three clients do not execute roof deposits, all roof deposits of other clients will be refunded. As can be seen from Figure 12b, since client 18 should make ladder deposits first among the three leaving clients, when client 18 discontinues ladder deposits, all the remaining clients will no longer perform ladder deposits and wait for the refund of their respective roof deposits. From Figure 12c, it can be seen that since client 8 performs the acknowledgment step first among the three leaving clients, after client 8 leaves with useful information, their ladder deposits will be used to compensate for the loss of client 1–client 7. Although client 14 and client 18 also leave midway, they do not obtain additional useful information, so all their deposits are returned. The fairness of FL is still guaranteed when multiple clients leave in the middle of the process.

To demonstrate the effect of including blockchain on training time, we give Figures 13 and 14.

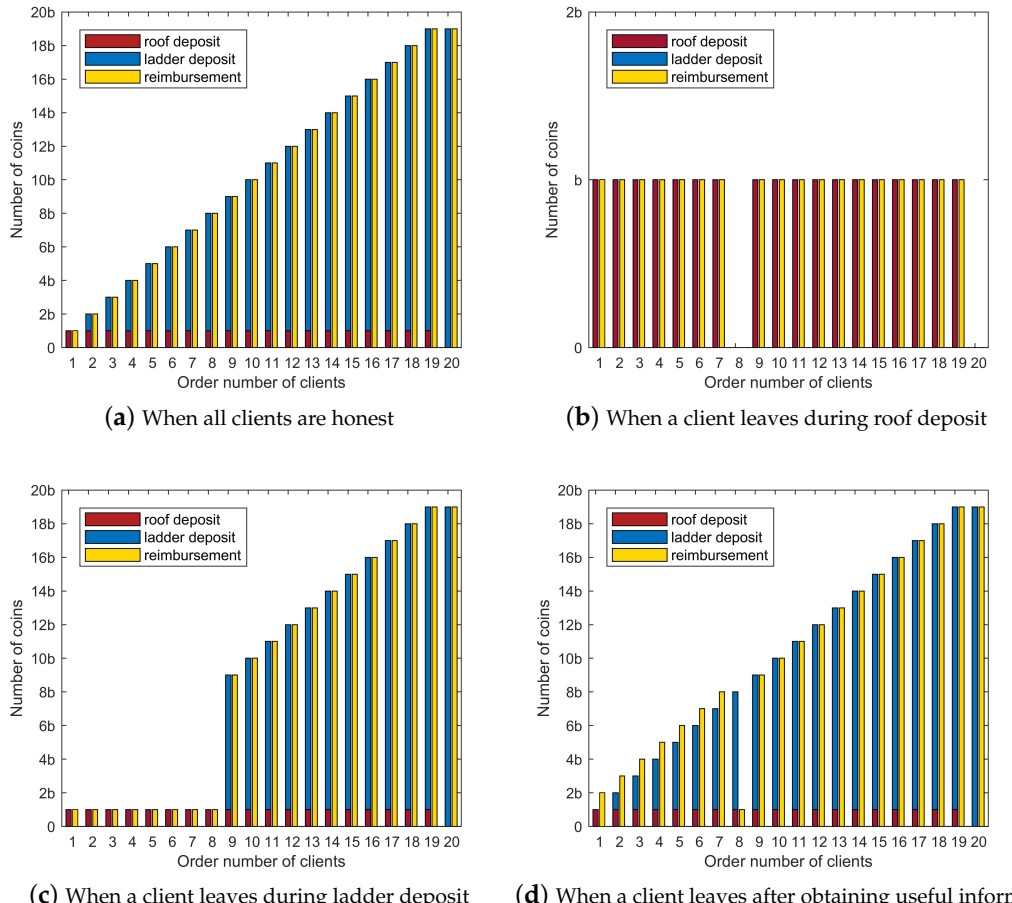

**Figure 11.** The circulation of digital currencies when there are 20 clients participating in FL.

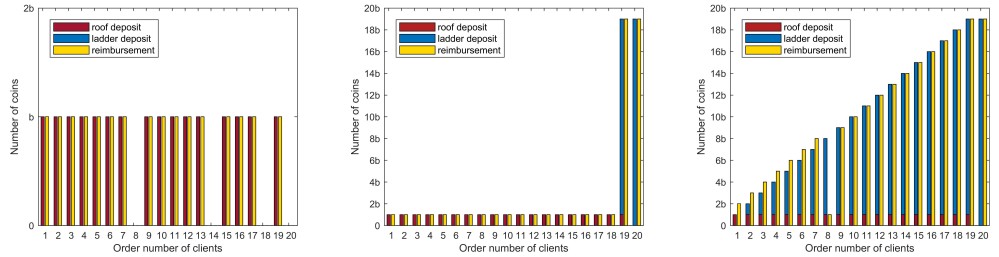

**Figure 12.** The circulation of digital currencies when there are 20 clients participating in FL and three leave halfway.

It can be seen from Figures 13 and 14 that including blockchain will take a little longer than excluding blockchain experiments with the MNIST and CIFAR10 data sets in both the IID and non-IID cases. If blockchain is not included, the time consumption mainly consists of the models' local training time, model transmission, model validation, and model aggregation time. In the case of including blockchain, the transaction confirmation time for paying deposits and deposit acknowledgment is increased, in which the confirmation operation of paying deposits and the models' local training can be executed in parallel. So, the increased time consumption mainly lies in the extra time for the confirmation operation of paying deposits over local training and the confirmation time of deposit acknowledgment. In the experiment, we use Ethereum 2.0,

which has a transaction confirmation time of 12–14 s. If a blockchain with a shorter consensus time such as Fabric is used, it can reduce the time consumption even more.

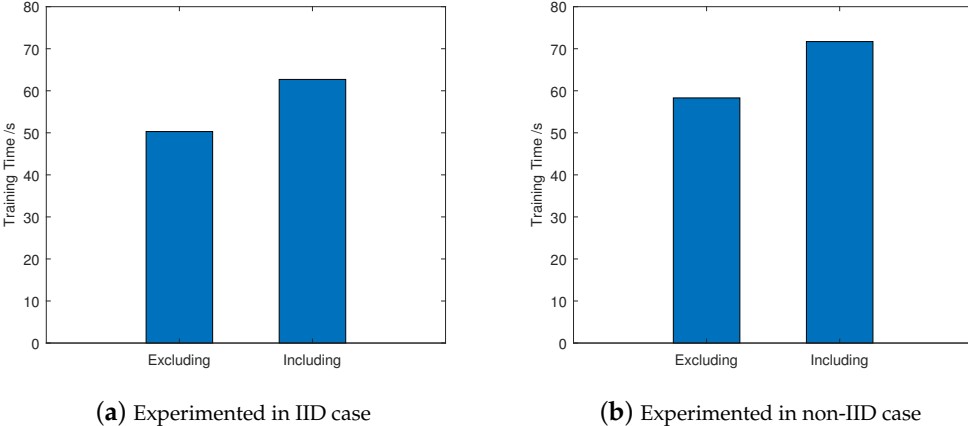

(**a**) Experimented in IID case    (**b**) Experimented in non-IID case

**Figure 13.** Comparisons of training time including and excluding blockchain of FL for MNIST data set in IID and non-IID cases.

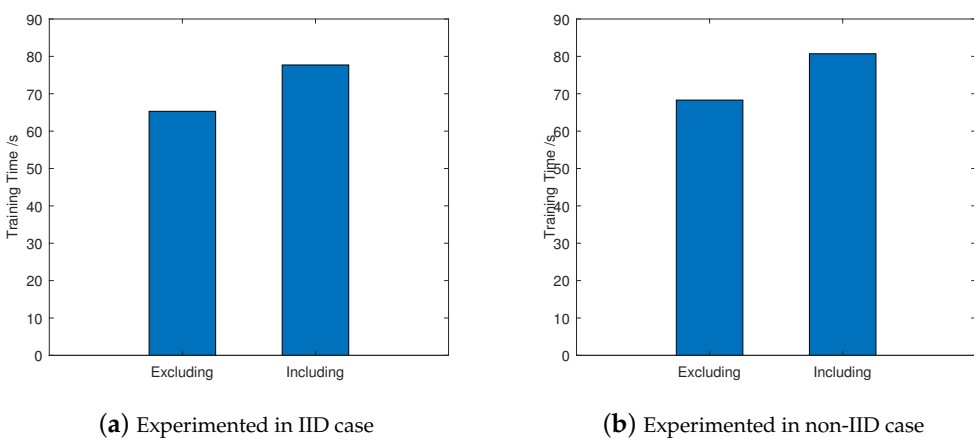

(**a**) Experimented in IID case    (**b**) Experimented in non-IID case

**Figure 14.** Comparisons of training time including and excluding blockchain of FL for CIFAR10 data set in IID and non-IID cases.

## 6. Conclusions

In this paper, we propose a ring-architecture-based fair federated learning framework RingFFL. The penalty mechanism is designed to ensure the fairness of FL in the case that the clients launch an escape attack during the FL process. That is, either all clients obtain the FL model, or clients who have suffered losses will be compensated with the digital currencies from the clients that launched escape attacks. Specifically, each client trains a local model based on its data and then transmits the local model parameters to other clients according to the rules designed in this paper. When dishonest clients obtain other clients' model parameters and abort, the honest clients who do not obtain the models will obtain a certain amount of digital currencies provided by dishonest clients using blockchain as a compensation, thus achieving the fairness of FL.

In the future, we plan to study the fairness of FL in the presence of more attacks. For example, when considering a poisoning attack in which the corrupted client manipulates the distribution of training data by inserting carefully crafted samples in the training set in order to change the model behavior and degrade the model performance; how to achieve fairness while guaranteeing the performance of FL is a question worth studying. In addition, taking asynchronous FL into account and investigating the case where there are clients who can join the training at any time may also be an interesting future direction.

**Author Contributions:** Methodology, L.H., X.H. and D.L.; supervision, Y.Z.; writing—original draft, L.H.; writing—review and editing, X.H., D.L. and Y.Z. All authors have read and agreed to the published version of the manuscript.

**Funding:** This research was funded by the "National Key Research and Development Program of China under Grant (No. 2020YFE0200500)".

**Data Availability Statement:** Data that support the findings of this study is available from the corresponding author upon reasonable request.

**Acknowledgments:** This work is supported by the National Key Research and Development Program of China under Grant (No.2020YFE0200500). The authors gratefully appreciate the anonymous reviewers for their valuable comments.

**Conflicts of Interest:** The authors declare no conflict of interest.

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
