# Peer review of "RingFFL: A Ring-Architecture-Based Fair Federated Learning Framework"

_futureinternet, doi:10.3390/fi15020068_

Round 1
Reviewer 1 Report
The authors propose a Federated Learning framework to penalize malicious clients by making them pay a deposit in advance according to the rules and record it on the blockchain.
The issue is essential in general; client selection and security are significant challenges in FL, so the paper addresses essential problems.
The idea is interesting even if, in my opinion, relying on research’s solutions to economic aspects makes me a lit bit worried. However, from a technical point of view, the key idea sounds useful and well-presented. Good presentation and the results.
However, a few points:
- What about honest clients that cannot pay the deposits or cannot do so?
- What is the impact of including blockchain aspects in training time?
- What about honest clients who left the architecture due to technical reasons?
- A strong improvement of related work should be addressed to report a brief introduction of all challenges of FL( https://doi.org/10.48550/arXiv.1905.10497, https://doi.org/10.1016/j.eswa.2021.116109, https://doi.org/10.48550/arXiv.2006.07242, https://doi.org/10.48550/arXiv.1905.06641, http://dx.doi.org/10.2139/ssrn.3696609, https://doi.org/10.1016/j.future.2022.06.006, https://doi.org/10.1016/j.ins.2022.11.126, and more...)
- Is the architecture suitable for mobile devices?
- A graphical view of the Penalty mechanism in RingFFL will help to understand the process better.
- Do the authors try to motivate the impact of NO-IID in Figure 5?
- Is there a relationship between dishonest clients and no-IDD?
- Please provide the future works.
Reviewer 2 Report
The authors should restructure the abstract to entail the following structure: Background, objectives, Materials and methods, results, conclusion, and recommendations.
However, it is not clear in this paper what the exact problem is.
Overall, the English language and presentation style should be improved significantly. There contained a lot of grammatical and punctuation errors, and typos.
The methodology should be enhanced to show the paper’s contribution rather than applying the existing tools.
The authors are advised to discuss further the data preprocessing and refinement in this work.
The primary contribution in comparison with competing methods should be highlighted in the introduction section to describe the novelty of the work.
Discuss the Limitations of the current work and future scope.
Reviewer 3 Report
The authors proposed a fair federated learning framework for managing fairness in federated learning. The authors are chosen a very good problem, but the contribution needs to be presented better. Fairness and security are the main problems in federated learning. The following comments may help to improve the quality of the paper.
i. Include more information about fairness in federated learning and present the issues properly.
ii. The Adversary model (3.2) problem needs to be presented better.
iii. Aggregation part needs to be described in the proposed model.
iv. What conditions or scenarios are considered for Normal situations and Abnormal situations for implementations?
v. Figure 7 to 10 implementation number of clients is very less. Increase the number of clients and re-do the experiments. Otherwise, give the reasons why these five clients are sufficient.
vi. Include the future direction of the research in the conclusion part.
Round 2
Reviewer 1 Report
The authors have successfully addressed all reviewers' concerns
Reviewer 2 Report
All my comments have been carefully attended to.
Reviewer 3 Report
The authors are addressed all my comments in the manuscript.